# Association of Chronic Periodontitis with Migraine in a Korean Adult Population: A Nationwide Nested Case-Control Study

**DOI:** 10.3390/healthcare13172123

**Published:** 2025-08-26

**Authors:** Joon Ho Song, Hyuntaek Rim, In Bok Chang, Hyo Geun Choi, Jee Hye Wee, Mi Jung Kwon, Ho Suk Kang, Ji Hee Kim

**Affiliations:** 1Department of Neurosurgery, Hallym University College of Medicine, Anyang 14068, Republic of Korea; song@hallym.or.kr (J.H.S.); coldwings@naver.com (H.R.); nscib71@hanmail.net (I.B.C.); 2Department of Otorhinolaryngology-Head and Neck Surgery, Suseoseoul ENT Clinic, Seoul 06349, Republic of Korea; pupen@naver.com; 3MD Analytics, Seoul 06349, Republic of Korea; 4Department of Otorhinolaryngology-Head and Neck Surgery, Hallym University College of Medicine, Anyang 14068, Republic of Korea; weejh07@hanmail.net; 5Department of Pathology, Hallym University College of Medicine, Anyang 14068, Republic of Korea; mulank@hanmail.net; 6Division of Gastroenterology, Department of Internal Medicine, Hallym University College of Medicine, Anyang 14068, Republic of Korea; hskang76@hallym.or.kr

**Keywords:** chronic periodontitis, inflammation, migraine, neurovascular mechanism, oral health

## Abstract

**Background:** Migraine and chronic periodontitis are prevalent conditions that may share common inflammatory and neurovascular pathways. Growing evidence suggests an association between periodontal inflammation and migraine, yet large-scale population-based studies are limited. **Objective:** To investigate the association between chronic periodontitis and the occurrence of migraine using a nested case-control design in a nationally representative Korean adult cohort. **Methods:** This study utilized data from the Korean National Health Insurance Service-Health Screening Cohort (2002–2019). A total of 43,359 individuals diagnosed with migraine (ICD-10: G43) were matched 1:4 by age, sex, income, and residence with 173,436 controls. Chronic periodontitis was identified using ICD-10 code K053. Conditional logistic regression was used to estimate adjusted odds ratios (ORs) and 95% confidence intervals (CIs), adjusting for demographic, behavioral, and clinical covariates. **Results:** A significant association was observed between chronic periodontitis and migraine. Individuals with at least one diagnosis of periodontitis within one year prior to migraine onset had increased odds of migraine (adjusted OR = 1.10, 95% CI: 1.08–1.13). Similar associations were observed for two diagnoses within one year (OR = 1.05; 95% CI: 1.01–1.09) and one diagnosis within two years (OR = 1.10; 95% CI: 1.08–1.13). No association was found with three or more diagnoses in one year. Subgroup analyses confirmed consistent associations across migraine subtypes and demographic strata. **Conclusions:** This study demonstrated a statistically significant association between chronic periodontitis and migraine, suggesting a potential shared inflammatory or neurovascular mechanism. Recognizing periodontal disease as a modifiable factor may offer new insights into migraine prevention and management. Further longitudinal and interventional studies are warranted to establish causality.

## 1. Introduction

Migraine is a prevalent and disabling neurological disorder experienced by approximately 15.1% of the world’s population [1]. It is typically characterized by recurrent episodes of moderate to severe headache, usually unilateral and pulsatile in nature, and frequently accompanied by nausea, photophobia, and phonophobia. These symptoms are generally exacerbated by routine physical activity. The pathogenesis of migraine involves a multifactorial interplay of environmental, genetic, and hormonal factors, with a notably higher prevalence among women [2,3,4]. Current research into migraine pathophysiology centers on two main areas: neurovascular dysregulation—including dysfunction of the trigeminovascular system and abnormal activation of brainstem nuclei—and neuroinflammatory processes, such as activation of microglia and mast cells, and elevated levels of calcitonin gene-related peptide (CGRP) and proinflammatory cytokines (e.g., IL-1β, IL-6, and TNF- α), which contribute to peripheral and central sensitization [1,5,6].

Periodontal diseases span a spectrum from reversible gingivitis to destructive periodontitis, including aggressive forms. Among these, chronic periodontitis is the most prevalent and clinically significant phenotype, defined as a destructive inflammatory disease of the supporting structures of the teeth. It arises from the host’s immune response to bacterial biofilm that accumulates on the non-shedding surfaces of the oral cavity [7]. Local inflammatory responses in periodontitis are characterized by the release of cytokines (IL-1β, IL-6, TNF- α), prostaglandins, and matrix metalloproteinases (MMPs) from resident and infiltrating immune cells, leading to tissue destruction and alveolar bone resorption [8]. Importantly, these local mediators can disseminate systemically, contributing to low-grade inflammation, endothelial dysfunction, and increased oxidative stress [9,10], which may in turn influence distant organs, including the central nervous system.

Historically regarded as a localized oral disease, growing evidence now indicates that chronic periodontitis is linked to systemic inflammation and may contribute to the pathogenesis of extra-oral disease. Numerous epidemiological and mechanistic studies have documented its association with systemic conditions, including cardiovascular disease, diabetes mellitus, rheumatoid arthritis, osteoporosis, and adverse pregnancy outcomes [11,12,13]. The etiology of periodontitis reflects a dysbiotic dental biofilm interacting with susceptible host immune response, modulated by oral hygiene behaviors, occlusal/malocclusion-related factors, genetic predisposition, and systemic conditions such as smoking and diabetes. Contemporary reviews emphasize this multifactorial framework and its clinical implications [14,15].

Of particular relevance to the present study, periodontitis has been implicated in several neurological disorders. Systemic reviews and meta-analyses have reported significant associations between chronic periodontitis and ischemic stroke [16], Alzheimer’s disease [17], Parkinson’s disease [18], and migraine [19,20,21]. These associations may be explained by shared inflammatory and vascular pathways, in which elevated systemic cytokines and CGRP from periodontal inflammation could exacerbate neurovascular dysfunction in migraine. Moreover, recent genetic studies suggest overlapping susceptibility loci between periodontitis and certain neurological diseases, supporting the hypothesis of shared pathophysiological mechanisms [22].

Although periodontitis has been linked to several neurological conditions, evidence specific to migraine remains limited, with most studies being small, cross-sectional, or clinic-based. Large population-based analyses with careful control of confounding are scarce, underscoring the need for the present investigation. Clarifying this relationship could yield clinically relevant insights and inform integrated care, whereby neurologists consider periodontal assessment or referral in migraine management, and public-health programs incorporate oral health promotion for populations with high migraine burden. We hypothesized that recent chronic periodontitis would be independently associated with higher odds of migraine, and that this association would be consistent across migraine subtypes and demographic strata.

Therefore, the objective of this study is to comprehensively examine the association between chronic periodontitis and the occurrence of migraine, and to determine whether recent diagnoses of periodontitis are associated with increased odds of migraine, using data from a large-scale nationally representative Korean adult cohort. Our study focuses specifically on chronic periodontitis (ICD-10 K05.3), as this was the only periodontal phenotype consistently encoded in the claims database; other periodontal entities could not be ascertained.

## 2. Methods

### 2.1. Data Source and Study Population

This study was approved by the Ethics Committee of Hallym University (23 October 2019). The Institutional Review Board waived the requirement for written informed consent because the analysis used fully anonymized secondary data from the Korean National Health Insurance Service (NHIS)-Health Screening Cohort, which contains no personal identifiers and poses minimal risk to participants. According to the guidelines of the Hallym University Ethics Committee and national regulations, studies using de-identified data without direct contact or intervention are eligible for exemption from informed consent.

The data for this nested case-control study were retrieved from the Korean NHIS-Health Screening Cohort data, consisting of 514,866 participants and 895,300,177 medical claim codes from 2002 to 2019. The nested case-control design was chosen to optimize analytic efficiency and reduce computational burden given the very large sample size, while still enabling detailed adjustment for confounding variables. To minimize any potential reduction in statistical power from sampling controls, a 1:4 matching strategy was applied, and conditional logistic regression was performed for analysis.

The migraine group included patients with at least two diagnoses of migraine (ICD-10 codes: G43) between 2002 and 2019 (n = 54,877). Two subtypes were defined: migraine with aura (G431) and migraine without aura (G430). Individuals with only one migraine diagnosis code were excluded because, in claims data, a single diagnosis often represents a provisional or rule-out diagnosis rather than a confirmed case. This approach was intended to enhance diagnostic accuracy and reduce misclassification. Individuals diagnosed with migraine in 2002 and 2003 were excluded to ensure that only new diagnoses were considered (n = 11,510). Individuals with missing data were also excluded (n = 8).

The control group was selected from participants not included in the migraine group (n = 459,989). Individuals with only one diagnosis of migraine between 2002 and 2019 were excluded (n = 50,307). Four control participants per migraine participant were randomly selected, matched by age, sex, income, and region of residence. The index date was defined as the date of the first migraine diagnosis for the case group and assigned accordingly for each matched control. Unmatched individuals were excluded (n = 236,246). The final analysis included 43,359 migraine participants and 173,436 control participants (Figure 1).

### 2.2. Identification of Chronic Periodontitis and Confounders

Chronic periodontitis was defined using ICD-10 codes K053 [23], based on the NHIS claims database. Although detailed clinical parameters such as probing death or attachment loss were not available, the NHIS database applies strict quality control processes and periodic audits to enhance diagnostic accuracy. Previous validation studies of NHIS claims data have reported acceptable reliability for periodontal diagnoses. Nevertheless, potential misclassification cannot be fully excluded, and this limitation is addressed in the Discussion Section. Participants were divided into ten age groups in 5-year intervals starting from age 40. Income was categorized into five levels (class 1: lowest income to class 5: highest income). Region of residence was classified as urban (including Seoul, Busan, Daegu, Incheon, Gwangju, Daejeon, and Ulsan) or rural (comprising Gyeonggi, Gangwon, Chungcheongbuk, Chungcheongnam, Jeollabuk, Jeollanam, Gyeongsangbuk, Gyeongsangnam, and Jeju). Smoking status was categorized as nonsmoker, past smoker, or current smoker. Alcohol consumption was categorized as less than once per week or once or more per week. Obesity was assessed using body mass index (BMI, kg/m^2^) and classified per the Asia-Pacific Western Pacific Regional Office (WPRO) 2000 standards: underweight (<18.5), normal (18.5–23), overweight (23–24.9), obese I (25–29.9), and obese II (≥30). Clinical measurements included systolic and diastolic blood pressure (mmHg), fasting blood glucose (mg/dL), and total cholesterol (mg/dL). The Charlson comorbidity index (CCI) was used to assess comorbid disease burden.

### 2.3. Statistical Analysis

The large sample size in this study was determined by the nature of the nationwide cohort, which included all eligible participants during the study period. As such, a priori sample size calculation or power analysis was not applicable; instead, we leveraged the entire dataset to maximize statistical power and representativeness. Demographics between the migraine and control groups were compared using standardized differences. The number of chronic periodontitis cases one or two years prior to the index date was described. Conditional logistic regression was applied to estimate odds ratio (OR) and 95% confidence interval (CI) for the association between chronic periodontitis and migraine. Model 1 adjusted for smoking status, alcohol use, obesity, and CCI score. Model 2 further adjusted for systolic and diastolic blood pressure (SBP and DBP), total cholesterol, and fasting blood glucose. Chronic periodontitis was assessed as follows: at least once, at least twice, and at least three times within one year, and at least once within two years. Subgroup analyses were performed based on migraine subtype and baseline characteristics. All analyses used SAS version 9.4 (SAS Institute Inc., Cary, NC, USA), with a two-tailed significance level of 0.05.

## 3. Results

Among 43,359 individuals with migraine and 173,436 controls, both groups were matched for sex, age, income, and region of residence (standardized difference = 0). The migraine group had slightly lower smoking and alcohol consumption rates and lower DBP but higher total cholesterol and CCI scores. SBP and fasting glucose levels were similar between groups. The number of periodontitis cases was higher in the migraine group both within one year (1.34%) and two years (2.08%) prior to the index date compared with the control group (1.28% and 1.99%, respectively, Table 1).

An increased probability of migraine was associated with at least one diagnosis of periodontitis within one year (OR in model 2 = 1.10, 95% CI = 1.08–1.13), two diagnoses within one year (OR in model 2 = 1.05, 95% CI = 1.01–1.09), and one diagnosis within two years (OR in model 2 = 1.10, 95% CI = 1.08–1.13). No association was found for three or more diagnoses within one year (OR in model 2 = 1.02, 95% CI = 0.97–1.06, Table 2). These associations remained consistent across migraine subtypes, except that two diagnoses within one year were not significantly associated with migraine with aura (Table 3 and Table 4).

Stratification by age, sex, income, region of residence, smoking status, alcohol consumption, BP, fasting blood glucose level, total cholesterol level, and CCI scores did not alter the estimates of migraine probability (Appendix A). This was also apparent when classifying individuals into migraine with aura or migraine without aura (Appendix A).

## 4. Discussion

This large-scale nested case-control study utilizing data from the Korean national health insurance service demonstrated a statistically significant association between chronic periodontitis and the occurrence of migraine. Even after controlling for a wide array of potential confounding variables, including demographic and lifestyle factors, comorbidities, and metabolic profiles, individuals with a recent diagnosis of chronic periodontitis were more likely to experience migraine episodes. These results support and extend findings from previous observational and case-control studies that suggest a bidirectional association between periodontal inflammation and neurological disorders such as migraine [24]. Recent meta-analyses further corroborate this link, showing that individuals with chronic migraine have more than double the odds of also having periodontitis compared with controls [25,26].

The biologic mechanisms underlying this association are likely multifactorial, involving chronic low-grade systemic inflammation, neurovascular dysregulation, and shared molecular mediators. Periodontitis is characterized by a persistent immune inflammatory response to bacterial biofilms, leading to systemic dissemination of cytokines such as tumor necrosis factor (TNF)-α, IL-1β, IL-6, and prostaglandins. These inflammatory mediators are capable of crossing the blood–brain barrier, where they may activate microglia and modulate the trigeminovascular system, an established pathway in migraine pathophysiology [14,27]. Elevated systemic inflammation from periodontitis could thus lower the threshold for migraine attacks or exacerbate migraine chronification. In addition, recent investigations have identified elevated serum levels of CGRP and procalcitonin in patients with both chronic periodontitis and chronic migraine, suggesting a shared inflammatory axis [28]. Leira et al. demonstrated that individuals with both conditions exhibited significantly higher serum procalcitonin levels than those with either condition alone or healthy controls, suggesting an overlapping inflammatory axis [28]. CGRP, a potent neuropeptide involved in vasodilation and nociception, has been well-established as a central mediator in migraine, and its upregulation in periodontal disease may provide a plausible biological pathway linking the two diseases. Moreover, recent research has highlighted leptin, an adipocytokine involved in immune modulation and metabolic regulation, as another potential biomarker linking these disorders. Elevated leptin levels, which have been observed in both migraine and periodontitis patients, may contribute to enhanced systemic inflammation and vascular dysregulation, providing a plausible mechanistic pathway. A novel layer of evidence comes from a two-sample Mendelian randomization study, which demonstrated a significant genetic predisposition linking periodontitis and migraine [29]. This suggests that shared genetic pathways, potentially involving immune regulatory genes and inflammatory cascades, may contribute to the co-occurrence of these disorders. These findings support our initial hypothesis that chronic periodontitis and migraine may share inflammatory and neurovascular mechanisms.

Interestingly, our study found that the association between periodontitis and migraine was most prominent among individuals with one or two diagnoses of chronic periodontitis within the prior one to two years. In contrast, no significant association was found for those with three or more diagnoses of chronic periodontitis within a single year. This paradoxical observation may be explained by several hypotheses. More frequent dental visits for chronic periodontitis might lead to more timely and effective treatment and improved periodontal health, thereby potentially attenuating systemic inflammation and mitigating the risk of migraine onset. Alternatively, it may reflect behavioral differences; those with severe periodontitis might also differ in health-seeking behavior or medication use, potentially influencing diagnosis coding or treatment intensity.

Our subgroup analyses further revealed that the association persisted across various demographic and clinical strata, including both migraine with and without aura. These findings align with multicenter studies such as the Spanish cross-sectional survey, which found a higher prevalence of self-reported periodontitis in chronic versus episodic migraineurs, even after adjusting for lifestyle and psychosocial variables [30]. Furthermore, recent investigations highlight that oral–systemic interactions may not be limited to periodontitis alone. Disorders such as temporomandibular dysfunction and facial myofascial pain have also been implicated in migraine development, reinforcing the notion that orofacial inflammatory or nociceptive inputs can influence central pain modulation [31].

However, several limitations should be acknowledged. First, both migraine and chronic periodontitis were identified using ICD-10 codes from administrative claims, which may introduce misclassification. However, NHIS diagnoses are regularly audited, and previous validation studies have reported acceptable accuracy. Second, as an observational study, causality cannot be inferred, and residual confounding (e.g., stress, sleep quality, diet, oral hygiene) may remain. Third, the structure of the claims database does not provide clinical indices (e.g., probing depth), which limits clinical granularity. Prospective studies are warranted to further investigate the temporal and causal relationship.

Despite these limitations, our study has notable strengths. It includes a large nationally representative cohort; thorough adjustment for confounders; and consistency of results across migraine subtypes and demographic strata. To our knowledge, this is among the largest investigations to date examining the link between periodontitis and migraine using real-world healthcare data.

From a clinical perspective, our findings suggest that periodontal health may play a role in migraine pathogenesis. We recommend that clinicians consider periodontal evaluation in migraine patients as part of holistic care. Interdisciplinary management, involving collaboration between neurologists and dental specialists, may enhance patient outcome. At a public health level, strategies promoting oral health may contribute to reducing migraine burden. Future studies should specifically investigate whether periodontal treatment reduces migraine frequency or severity in randomized controlled trials. Biomarker-based studies could clarify inflammatory pathways linking the two conditions. Additionally, replication in diverse populations and age groups is needed to test generalizability. While this study provides evidence of a significant association between chronic periodontitis and migraine in a Korean adult population, it remains uncertain whether these findings can be generalized across other ethnic groups. Genetic background, environmental exposures, lifestyle behaviors, and socio-cultural factors may all influence both periodontal disease and migraine susceptibility. Therefore, future research should include ethnically and geographically diverse populations to confirm whether the observed association is consistent across different contexts and to strengthen the external validity of these findings.

## 5. Conclusions

In conclusion, this nationwide study identified a significant association between chronic periodontitis and migraine. Clinically, incorporating periodontal health assessment into migraine management could be beneficial. Future interdisciplinary approaches, such as neurologic–dental collaborations and public health interventions promoting oral health, may provide new opportunities for migraine prevention and management.

## Figures and Tables

**Figure 1 healthcare-13-02123-f001:**
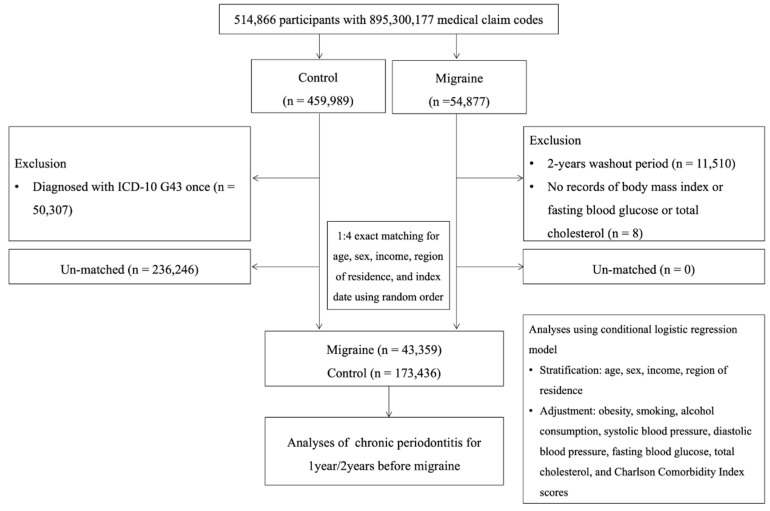
Flowchart illustrating the inclusion and exclusion criteria for this study. A total of 514,866 participants were enrolled in this study, including 43,359 subjects with migraine and 173,436 control subjects matched for age, sex, income, and region of residence.

**Table 1 healthcare-13-02123-t001:** General characteristics of participants.

Characteristics	Total Participants		
	Migraine	Control	Standardized Difference
Age (years old) (n, %)			0
40–44	934 (2.15)	3736 (2.15)	
45–49	4641 (10.70)	18,564 (10.70)	
50–54	6815 (15.72)	27,260 (15.72)	
55–59	7283 (16.80)	29,132 (16.80)	
60–64	6608 (15.24)	26,432 (15.24)	
65–69	6589 (15.20)	26,356 (15.20)	
70–74	5349 (12.34)	21,396 (12.34)	
75–79	3290 (7.59)	13,160 (7.59)	
80–84	1423 (3.28)	5692 (3.28)	
85+	427 (0.98)	1708 (0.98)	
Sex (n, %)			0
Male	14,717 (33.94)	58,868 (33.94)	
Female	28,642 (66.06)	114,568 (66.06)	
Income (n, %)			0
1 (lowest)	8026 (18.51)	32,104 (18.51)	
2	6113 (14.10)	24,452 (14.10)	
3	7035 (16.23)	28,140 (16.23)	
4	9060 (20.90)	36,240 (20.90)	
5 (highest)	13,125 (30.27)	52,500 (30.27)	
Region of residence (n, %)			0
Urban	16,999 (39.21)	67,996 (39.21)	
Rural	26,360 (60.79)	105,440 (60.79)	
Obesity ^†^ (n, %)			0.07
Underweight	1046 (2.41)	4411 (2.54)	
Normal	15,442 (35.61)	62,425 (35.99)	
Overweight	11,743 (27.08)	46,280 (26.68)	
Obese I	13,760 (31.74)	54,495 (31.42)	
Obese II	1368 (3.16)	5825 (3.36)	
Smoking status (n, %)			0.03
Nonsmoker	34,538 (79.66)	136,296 (78.59)	
Past smoker	2541 (5.86)	10,243 (5.91)	
Current smoker	6280 (14.48)	26,897 (15.51)	
Alcohol consumption (n, %)			0.04
<1 time a week	35,332 (81.49)	138,775 (80.02)	
≥1 time a week	8027 (18.51)	34,661 (19.98)	
Systolic blood pressure (n, %)			0.06
<120 mmHg	13,930 (32.13)	53,541 (30.87)	
120–139 mmHg	20,775 (47.91)	81,965 (47.26)	
≥140 mmHg	8654 (19.96)	37,930 (21.87)	
Diastolic blood pressure (n, %)			0.04
<80 mmHg	21,324 (49.18)	83,804 (48.32)	
80–89 mmHg	15,278 (35.24)	59,928 (34.55)	
≥90 mmHg	6757 (15.58)	29,704 (17.13)	
Fasting blood glucose (n, %)			0.07
<100 mg/dL	28,602 (65.97)	109,716 (63.26)	
100–125 mg/dL	11,581 (26.71)	47,996 (27.67)	
≥126 mg/dL	3176 (7.32)	15,724 (9.07)	
Total cholesterol (n, %)			0
<200 mg/dL	22,632 (52.20)	90,891 (52.41)	
200–239 mg/dL	14,471 (33.37)	57,592 (33.21)	
≥240 mg/dL	6256 (14.43)	24,953 (14.39)	
CCI score (n, %)			0.04
0	23,948 (55.23)	105,933 (61.08)	
1	8581 (19.79)	27,566 (15.89)	
≥2	10,830 (24.98)	39,937 (23.03)	
The number of chronic periodontitis cases (mean, standard deviation)			
within 1 year	0.50 (1.34)	0.47 (1.28)	0.02
within 2 years	0.95 (2.08)	0.90 (1.99)	0.02

CCI—Charlson comorbidity index. ^†^ Obesity (BMI—body mass index, kg/m^2^) was categorized as <18.5 (underweight), ≥18.5 to <23 (normal), ≥23 to <25 (overweight), ≥25 to <30 (obese I), and ≥30 (obese II).

**Table 2 healthcare-13-02123-t002:** Crude and adjusted odds ratios for the association between chronic periodontitis and migraine.

Characteristics	No. of Case (Exposure/Total, %)	No. of Control (Exposure/Total, %)	Odds Ratios for Migraine (95% Confidence Interval)					
			Crude ^†^	*p*-Value	Model 1 ^†‡^	*p*-Value	Model 2 ^†§^	*p*-Value
Migraine (n = 216,795)								
CP ≥ 1 (1 year)	9786/43,359 (22.6%)	36,379/173,436 (21.0%)	1.10 (1.07–1.13)	<0.001 *	1.11 (1.08–1.13)	<0.001 *	1.10 (1.08–1.13)	<0.001 *
CP ≥ 2 (1 year)	4716/43,359 (10.9%)	18,149/173,436 (10.5%)	1.04 (1.01–1.08)	0.012 *	1.05 (1.02–1.09)	0.005 *	1.05 (1.01–1.09)	0.005 *
CP ≥ 3 (1 year)	2584/43,359 (6.0%)	10,242/173,436 (5.9%)	1.01 (0.97–1.06)	0.667	1.02 (0.97–1.06)	0.473	1.02 (0.97–1.06)	0.476
CP ≥ 1 (2 years)	15,077/43,359 (34.8%)	56,850/173,436 (32.8%)	1.10 (1.07–1.12)	<0.001 *	1.10 (1.08–1.13)	<0.001 *	1.10 (1.08–1.13)	<0.001 *

CP—chronic periodontitis. * Conditional or unconditional logistic regression analysis, significance at *p* < 0.05. ^†^ Stratified model for age, sex, income, and geographic region. ^‡^ Model 1 was adjusted for smoking status, alcohol use, obesity, and CCI scores. ^§^ Model 2 was adjusted for model 1 plus total cholesterol, SBP, DBP, and fasting blood glucose.

**Table 3 healthcare-13-02123-t003:** Crude and adjusted odds ratios for the association between chronic periodontitis and migraine with aura.

Characteristics	No. of Case (Exposure/Total, %)	No. of Control (Exposure/Total, %)	Odds Ratios for Migraine with Aura (95% Confidence Interval)					
			Crude ^†^	*p*-Value	Model 1 ^†‡^	*p*-Value	Model 2 ^†§^	*p*-Value
Migraine with aura (n = 15,760)								
CP ≥ 1 (1 year)	688/3152 (21.8%)	2461/12,608 (19.5%)	1.15 (1.05–1.27)	0.004 *	1.17 (1.06–1.28)	0.002 *	1.16 (1.06–1.28)	0.002 *
CP ≥ 2 (1 year)	314/3152 (10.0%)	1216/12,608 (9.6%)	1.04 (0.91–1.18)	0.588	1.05 (0.92–1.20)	0.481	1.05 (0.92–1.19)	0.501
CP ≥ 3 (1 year)	151/3152 (4.8%)	659/12,608 (5.2%)	0.91 (0.76–1.09)	0.319	0.92 (0.77–1.10)	0.366	0.91 (0.76–1.10)	0.324
CP ≥ 1 (2 years)	1067/3152 (33.9%)	3836/12,608 (30.4%)	1.18 (1.08–1.28)	<0.001 *	1.19 (1.09–1.29)	<0.001 *	1.18 (1.09–1.29)	<0.001 *

CP—chronic periodontitis. * Conditional or unconditional logistic regression analysis, significance at *p* < 0.05. ^†^ Stratified model for age, sex, income, and geographic region. ^‡^ Model 1 was adjusted for smoking status, alcohol use, obesity, and CCI scores. ^§^ Model 2 was adjusted for model 1 plus total cholesterol, SBP, DBP, and fasting blood glucose.

**Table 4 healthcare-13-02123-t004:** Crude and adjusted odds ratios for the association between chronic periodontitis and migraine without aura.

Characteristics	No. of Case (Exposure/Total, %)	No. of Control (Exposure/Total, %)	Odds Ratios for Migraine Without Aura (95% Confidence Interval)					
			Crude ^†^	*p*-Value	Model 1 ^†‡^	*p*-Value	Model 2 ^†§^	*p*-Value
Migraine without aura (n = 201,035)								
(n = 15,760)								
CP ≥ 1 (1 year)	9098/40,207 (22.6%)	33,918/160,828 (21.1%)	1.10 (1.07–1.13)	<0.001 *	1.10 (1.07–1.13)	<0.001 *	1.10 (1.07–1.13)	<0.001 *
CP ≥ 2 (1 year)	4402/40,207 (11.0%)	16,933/160,828 (10.5%)	1.05 (1.01–1.08)	0.014 *	1.05 (1.01–1.09)	0.006 *	1.05 (1.01–1.09)	0.006 *
CP ≥ 3 (1 year)	2433/40,207 (6.1%)	9583/160,828 (6.0%)	1.02 (0.97–1.06)	0.482	1.02 (0.98–1.07)	0.335	1.02 (0.98–1.07)	0.33
CP ≥ 1 (2 years)	14,010/40,207 (34.8%)	53,014/160,828 (33.0%)	1.09 (1.06–1.12)	<0.001 *	1.10 (1.07–1.12)	<0.001 *	1.09 (1.07–1.12)	<0.001 *

CP—chronic periodontitis. * Conditional or unconditional logistic regression analysis, significance at *p* < 0.05. ^†^ Stratified model for age, sex, income, and geographic region. ^‡^ Model 1 was adjusted for smoking status, alcohol use, obesity, and CCI scores. ^§^ Model 2 was adjusted for model 1 plus total cholesterol, SBP, DBP, and fasting blood glucose.

## Data Availability

The data presented in this study are available on request from the corresponding author (kimjihee.ns@gmail.com).

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
