# Peer review of "Association of Chronic Periodontitis with Migraine in a Korean Adult Population: A Nationwide Nested Case-Control Study"

_healthcare, 2025, doi:10.3390/healthcare13172123_

Round 1
Reviewer 1 Report
Comments and Suggestions for Authors
The Abstract, emphatically states that migraine, which periodontitis is, is an inflammatory disorder. Increasing evidence suggests that migraine is a polygenic neurovascular disorder and its pathophysiology is not completely understood. I suggest to evaluate this affirmation.
Introduction: Very brief. I consider that it should be expanded by including more bibliography on the association of this entity with other systemic diseases, mainly neurological (e.g. systematic reviews)
Objective: I consider it must be more precise (“comprehensively examine” or to determine the …..”)
Methods: I suggest to explain why the Institutional Review Board waived the requirement for written informed consent. Generally, the Ethics Committee approves the request for exemption from informed consent documentation (e.g., minimal risk of harm)
To explain why the use of a nested case-control study that has the disadvantage of reduced precision and power due to sampling of controls.
In the paragraph of the selection of the control group appears the exclusion of individuals with only one diagnosis of migraine. I suggest to put it in the migraine group
Results:
Tables are too full of data. It is suggested to split them.
Author Response
Comment 1: The Abstract, emphatically states that migrainex, which periodontitis is, is an inflammatory disorder. Increasing evidence suggests that migraine is a polygenic neurovascular disorder and its pathophysiology is not completely understood. I suggest to evaluate this affirmation.
Response: Thank you for the constructive and informative comments. We agree that migraine is primarily recognized as a complex polygenic neurovascular disorder, and its pathophysiology is not yet fully elucidated. While both migraine and chronic periodontitis are associated with inflammatory processes, we acknowledge that it would be more accurate to avoid implying that migraine itself is strictly an inflammatory disorder. Therefore, we have modified the Abstract of the original manuscript as follows:
Previous abstract: “Migraine and chronic periodontitis are prevalent inflammatory disorders that may share common pathophysiological pathways.” Revised abstract (page 2, lines 18-19): “Migraine and chronic periodontitis are prevalent conditions that may share common inflammatory and neurovascular pathways.” |
Comment 2: Introduction: Very brief. I consider that it should be expanded by including more bibliography on the association of this entity with other systemic diseases, mainly neurological (e.g. systematic reviews). Objective: I consider it must be more precise (“comprehensively examine” or to determine the …..”).
Response: Thank you for valuable suggestion. We agree that the Introduction would benefit from a more comprehensive background on the association of chronic periodontitis with other systemic diseases, particularly neurological disorders. In response, we have expanded the Introduction to include additional references, including recent systematic reviews, that discuss the relationship between periodontitis and various systemic conditions such as ischemic stroke, Alzheimer’s disease, Parkinson’s disease, and migraine. Accordingly, we have revised it to clearly convey the aim of the study.
Previous introduction: “Although historically considered a localized oral condition, growing evidence now suggests that chronic periodontitis contributes to systemic inflammation and is associated with various systemic conditions, including ischemic stroke and Alzheimer’s disease [5-7]. Chronic periodontitis has also been linked to other systemic disorders, such as diabetes mellitus, cardiovascular diseases, rheumatoid arthritis, osteoporosis, and reproductive health issues in both men and women. Importantly, associations with neurodegenerative and neurological conditions–including Alzheimer’s, disease, Parkinson’s disease, and migraine–have also been identified [8]. The objective of this study is to comprehensively examine the association between chronic periodontitis and the occurrence of migraine in the Korean adult population, utilizing data from a large-scale national healthcare database.” Revised introduction (page 3, lines 76-111): “Historically regarded as a localized oral disease, growing evidence now indicates that chronic periodontitis is linked to systemic inflammation and may contribute to the pathogenesis of extra-oral disease. Numerous epidemiological and mechanistic studies have documented its association with systemic conditions, including cardiovascular disease, diabetes mellitus, rheumatoid arthritis, osteoporosis, and adverse pregnancy outcomes [12-14]. The etiology of periodontitis reflects a dysbiotic dental biofilm interacting with susceptible host immune response, modulated by oral-hygiene behaviors, occlusal/malocclusion-related factors, genetic predisposition, and systemic conditions such as smoking and diabetes. Contemporary reviews emphasize this multifactorial framework and its clinical implications [15,16]. Of particular relevance to the present study, periodontitis has been implicated in several neurological disorders. Systemic reviews and meta-analyses have reported significant associations between chronic periodontitis and ischemic stroke [17], Alzheimer’s disease [18], Parkinson’s disease [19], and migraine [20-22]. These associations may be explained by shared inflammatory and vascular pathways, in which elevated systemic cytokines and CGRP from periodontal inflammation could exacerbate neurovascular dysfunction in migraine. Moreover, recent genetic studies suggest overlapping susceptibility loci between periodontitis and certain neurological diseases, supporting the hypothesis of shared pathophysiological mechanisms [23]. Although periodontitis has been linked to several neurological conditions, evidence specific to migraine remains limited, with most studies being small, cross-sectional, or clinic-based. Large, population-based analyses with careful control of confounding are scarce, underscoring the need for the present investigation. Clarifying this relationship could yield clinically relevant insights and inform integrated care, whereby neurologist consider periodontal assessment or referral in migraine management, and public-health programs incorporate oral-health promotion for populations with high migraine burden. We hypothesized that recent chronic periodontitis would be independently associated with higher odds of migraine, and that this association would be consistent across migraine subtypes and demographic strata. Therefore, the objective of this study is to comprehensively examine the association between chronic periodontitis and the occurrence of migraine, and to determine whether recent diagnoses of periodontitis are associated with increased odds of migraine, using data from a large-scale, nationally representative Korean adult cohort. Our study focuses specifically on chronic periodontitis (ICD-10 K05.3), as this was the only periodontal phenotype consistently encoded in the claims database; other periodontal entities could not be ascertained.” |
Comment 3: Methods: I suggest to explain why the Institutional Review Board waived the requirement for written informed consent. Generally, the Ethics Committee approves the request for exemption from informed consent documentation (e.g., minimal risk of harm) To explain why the use of a nested case-control study that has the disadvantage of reduced precision and power due to sampling of controls.
In the paragraph of the selection of the control group appears the exclusion of individuals with only one diagnosis of migraine. I suggest to put it in the migraine group.
Response: We appreciate your insightful comment. 1. Waiver of written informed consent: We appreciate your point regarding the explanation for the Institutional Review Board (IRB) waiver. In this study, we used de-identified secondary data from the Korean National Health Insurance Service-Health Screening Cohort, which contains no personal identifiers and poses minimal risk to participants. According to the guidelines of the Hallym University Ethics Committee and national regulations, studies using anonymized data without direct contact or intervention are eligible for exemption from written informed consent. We have revised the Methods section to clarify this rationale.
Previous methods: “The Institutional Review Board waived the requirement for written informed consent. All procedures complied with the ethical standards and regulations set by the Hallym University Ethics Committee.” Revised methods (pages 3-4, lines 115-172): “The Institutional Review Board waived the requirement for written informed consent because the analysis used fully anonymized secondary data from the Korean National Health Insurance Service (NHIS)-Health Screening Cohort, which contains no personal identifiers and poses minimal risk to participants. According to the guidelines of the Hallym University Ethics Committee and national regulations, studies using de-identified data without direct contact or intervention are eligible for exemption from informed consent.”
2. Use of a nested case-control design: We acknowledge your concern about the reduced precision and power inherent to a nested case-control design due to sampling of controls. This design was chosen to optimize efficiency and reduce computational burden when analyzing a large-scale national cohort with over 500,000 participants and nearly 900 million medical claims. Furthermore, we used 1:4 matching and conditional logistic regression to mitigate potential loss of statistical power. This explanation has been added to the Methods section.
Previous methods: “The detailed information of the Korean National Health Insurance Service-Health Screening Cohort data was described elsewhere thoroughly.” Revised methods (page 4, lines 175-179): “The nested case-control design was chosen to optimize analytic efficiency and reduce computational burden given the very large sample size, while still enabling detailed adjustment for confounding variables. To minimize any potential reduction in statistical power from sampling controls, a 1:4 matching strategy was applied, and conditional logistic regression was performed for analysis.”
3. Exclusion of individuals with only one diagnosis of migraine: We agree that clarification is needed. Individuals with only one migraine diagnosis code were excluded to enhance diagnostic accuracy and minimize potential misclassification. In Korean claims data, a single diagnosis code may reflect a provisional or rule-out diagnosis rather than a confirmed case. For this reason, only individuals with at least two migraine diagnoses were classified into the migraine group. This rationale has been added to the description of the migraine group selection in the Methods section.
Revised methods (page 4, lines 182-185): “Individuals with only one migraine diagnosis code were excluded because, in claims data, a single diagnosis often represents a provisional or rule-out diagnosis rather than a confirmed case. This approach was intended to enhance diagnostic accuracy and reduce misclassification.” |
Comment 4: Results: Tables are too full of data. It is suggested to split them.
Response: Thank you for the valuable comments. We agree that splitting large tables will improve readability and clarity. We have divided Table 2 into three separate tables (Table 2 for all participants, Table 3 for migraine with aura, and Table 4 for migraine without aura). This restructuring enhances data presentation and allows for easier interpretation of the results. |
Reviewer 2 Report
Comments and Suggestions for Authors
The authors present a very interesting paper regarding the Association of Chronic Periodontitis with Migraine in A Korean Adult Population. The methods are appropriate and well described, and adequate details are provided to replicate the work. The discussion and conclusions are well balanced and adequately supported by the data. The paper is clearly written However, I recommend some minor revisions aimed at improving the clarity, scientific rigor, and overall quality of the manuscript.
Introduction:
The introduction could benefit from further expansion to provide a more comprehensive background for the study. While it offers a useful overview of migraine and chronic periodontitis, the pathophysiological aspects are only briefly addressed. We suggest including a more detailed discussion of the underlying mechanisms (e.g., neurovascular and neuroinflammatory processes in migraine; local and systemic inflammatory pathways in periodontitis) to enhance the scientific depth. In addition, the statement in lines 51–53 should be supported with an appropriate reference.
Discussion:
The discussion thoroughly addresses the main aspects of the topic, examining possible biological mechanisms behind the association between chronic periodontitis and migraine and citing relevant clinical and experimental evidence. However, it would be useful to highlight the need for studies including populations of different ethnic backgrounds, to assess whether the association holds across diverse genetic, environmental, and socio-cultural contexts and to strengthen the overall validity of the findings.
Author Response
The authors present a very interesting paper regarding the Association of Chronic Periodontitis with Migraine in A Korean Adult Population. The methods are appropriate and well described, and adequate details are provided to replicate the work. The discussion and conclusions are well balanced and adequately supported by the data. The paper is clearly written However, I recommend some minor revisions aimed at improving the clarity, scientific rigor, and overall quality of the manuscript.
Comment 1: Introduction: The introduction could benefit from further expansion to provide a more comprehensive background for the study. While it offers a useful overview of migraine and chronic periodontitis, the pathophysiological aspects are only briefly addressed. We suggest including a more detailed discussion of the underlying mechanisms (e.g., neurovascular and neuroinflammatory processes in migraine; local and systemic inflammatory pathways in periodontitis) to enhance the scientific depth. In addition, the statement in lines 51–53 should be supported with an appropriate reference.
Response: We sincerely thank the reviewer for this constructive suggestion. In response, we have expanded the Introduction to provide a more comprehensive discussion of the underlying pathophysiological mechanisms. Specifically, we have elaborated on: 1. Neurovascular and neuroinflammatory processes in migraine, including the role of the trigeminovascular system, microglial activation, and elevated inflammatory mediators such as CGRP, IL-1 , IL-6, and TNF- . 2. Local and systemic inflammatory pathways in chronic periodontitis, highlighting how proinflammatory cytokines, prostaglandins, and matrix metalloproteinases contribute to both local tissue destruction and systemic low-grade inflammation. 3. Shared mechanisms between migraine and periodontitis, including systemic dissemination of inflammatory mediators and possible genetic susceptibility. In addition, we have added appropriate references to support the statement in lines 51-53 regarding the systemic inflammatory impact of periodontitis. This revision enhances the scientific depth of the Introduction and provide a stronger rationale for the present study.
Previous introduction: “Current research into migraine pathophysiology centers on two main areas: the involvement of specific neurotransmitters and the contribution of inflammatory processes.” Revised introduction (page 2, lines 51–56): “Current research into migraine pathophysiology centers on two main areas: neurovascular dysregulation–including dysfunction of the trigeminovascular system and abnormal activation of brainstem nuclei–and neuroinflammatory processes, such as activation of microglia and mast cells, and elevated levels of calcitonin gene-related peptide (CGRP) and proinflammatory cytokines (e.g., IL-1ß, IL-6, and TNF- ), which contribute to peripheral and central sensitization [5-7].”
Previous introduction: “Chronic periodontitis, similarly, is a persistent and multifactorial inflammatory disease affecting the supporting structures of the teeth. It arises from the host’s immune response to bacterial biofilm that accumulates on the nonshedding surfaces of the oral cavity. Like migraine, chronic periodontitis is a significant contributor to the global burden of chronic diseases and poses a major public health concern. The disease processes slowly and painlessly, leading to periodontal attachment loss and alveolar bone resorption, which may ultimately result in tooth mobility and loss.” Revised introduction (pages 2-3, lines 57–75): “Periodontal diseases span a spectrum from reversible gingivitis to destructive periodontitis, including aggressive forms. Among these, chronic periodontitis is the most prevalent and clinically significant phenotype, defined as a destructive inflammatory disease of the supporting structures of the teeth. It arises from the host’s immune response to bacterial biofilm that accumulates on the nonshedding surfaces of the oral cavity [8]. Local inflammatory responses in periodontitis are characterized by the release of cytokines (IL-1ß, IL-6, TNF- ), prostaglandins, and matrix metalloproteinases (MMPs) from resident and infiltrating immune cells, leading to tissue destruction and alveolar bone resorption [9]. Importantly, these local mediators can disseminate systemically, contributing to low-grade inflammation, endothelial dysfunction, and increased oxidative stress [10,11], which may in turn influence distant organs, including the central nervous system.” |
Comment 2: Discussion: discussion thoroughly addresses the main aspects of the topic, examining possible biological mechanisms behind the association between chronic periodontitis and migraine and citing relevant clinical and experimental evidence. However, it would be useful to highlight the need for studies including populations of different ethnic backgrounds, to assess whether the association holds across diverse genetic, environmental, and socio-cultural contexts and to strengthen the overall validity of the findings.
Response: We appreciate the reviewer’s constructive and informative comments. We agree that assessing whether the observed association between chronic periodontitis and migraine is consistent across different ethnic background is important to strengthen the generalizability of our findings. In response, we have revised the Discussion section to highlight the need for further studies involving populations from diverse genetic, environmental, and sociocultural contexts. This addition underscores the importance of cross-ethnic investigations to validate our results and to determine whether unique population-specific factors may influence the relationship.
Revised discussion (page 10, lines 381–388): “While this study provides evidence of a significant association between chronic periodontitis and migraine in a Korean adult population, it remains uncertain whether these findings can be generalized across other ethnic groups. Genetic background, environmental exposures, lifestyle behaviors, and socio-cultural factors may all influence both periodontal disease and migraine susceptibility. Therefore, future research should include ethnically and geographically diverse populations to confirm whether the observed association is consistent across different contexts and to strengthen the external validity of these findings.” |
Reviewer 3 Report
Comments and Suggestions for Authors
Association of chronic periodontitis with migraine in a Korean adult population: A nationwide nested case-control study
Reviewer Report
This study was conducted using data from the Korean National Health Insurance Service and employed a nested case-control design, including 43,359 individuals with migraine and 173,436 matched controls. Individuals diagnosed with chronic periodontitis at least once within one year prior to migraine onset were found to have a significantly higher likelihood of developing migraine. This association remained largely consistent across analyses based on the number and timing of diagnoses. Similar results were also observed across different migraine subtypes and demographic groups. The authors concluded that chronic periodontitis may be a potential modifiable risk factor to consider in the prevention and management of migraine. Although the study addresses an interesting topic, there are significant scientific and structural shortcomings that require revision. Detailed comments are provided below:
Introduction
This section lacks or insufficiently addresses some essential elements that should be included in the introduction of a scientific study. A detailed evaluation is provided below in bullet points:
- The diversity of periodontal diseases (such as gingivitis, aggressive periodontitis, etc.) is not mentioned at all. Only chronic periodontitis is addressed, while the prevalence and systemic effects of other periodontal conditions are overlooked.
- The etiological factors of periodontitis, including dental factors (e.g., dental plaque, poor oral hygiene, malocclusion, dental biofilm) and non-dental factors (e.g., genetic predisposition, immune response, smoking, systemic diseases), are not explained. The etiological framework is inadequate. Please consider adding up-to-date literature on these aspects: https://doi.org/10.1186/s12903-025-06449-6 and https://doi.org/10.1016/j.ortho.2019.08.025
- The authors mention the association of chronic periodontitis with neurological diseases but do not clearly emphasize that this relationship has been insufficiently studied specifically in relation to migraine. Since this gap in the literature is not clearly stated, readers may not be sufficiently convinced about the importance of the study. In other words, the question "Why was this study conducted?" is not answered convincingly.
- The potential clinical contributions of the study (e.g., considering periodontal health in migraine management) are not described. There is a lack of emphasis on how this study could directly benefit clinicians or public health, which should be clearly stated.
- The study lacks a hypothesis. In academic studies, especially analytical observational research, it is expected that the research hypothesis is explicitly stated.
Methods
There are some important points missing or requiring clarification in Methods section. Detailed evaluation and recommendations are provided below:
- Definition of chronic periodontitis is based solely on the ICD-10 code, with no information provided regarding clinical diagnostic or measurement criteria or the accuracy of the diagnosis. This is especially important for registry-based studies concerning diagnostic reliability.
- Although a very large sample size was used, there is no explanation of how the sample size was calculated or any justification or power analysis supporting the need for such a large sample. This represents a methodological deficiency and should be addressed.
- While the data source (Korean National Health Insurance Service database) is large and comprehensive, no information is given about data quality, diagnostic accuracy, the timeliness of records, or potential errors. Additionally, the limitations and possible misclassifications inherent in using ICD-10 codes to identify diseases are not discussed. Such explanations regarding data reliability are important for methodological transparency.
- Given that this is a registry-based observational study, the limitations inherent to the study design (such as the inability to infer causality) should be acknowledged and discussed in Discussion section.
Results
Results section is prepared with methodological rigor, supported by both overall and subgroup analyses, and organized in a way that allows readers to easily follow the study’s findings. No obvious deficiencies requiring revision or additional explanation are observed.
Discussion
Discussion section requires improvement in some aspects. Below are my detailed evaluations and recommendations:
- Some conflicting views or alternative biological mechanisms in the relevant literature should be presented more balancedly. A clearer and more understandable explanation is needed.
- Although limitations such as reliance on ICD codes for diagnoses, inability to infer causality, and potential unmeasured confounders (stress, sleep, diet, oral hygiene) are mentioned, the limitations section should be more systematically and thoroughly outlined. For example, more explanation about structural limitations and potential errors in the data source should be added.
- The role of periodontal health in migraine pathogenesis is well emphasized and regarded as a modifiable risk factor in clinical practice. However, this contribution should be supported with more concrete recommendations. For instance, suggesting clinicians perform periodontal assessments or advocating for multidisciplinary approaches. Additionally, interdisciplinary patient management and public health perspectives could be expanded further.
- For future studies, the need for prospective and interventional research is stated. However, the recommendations could be more specific—for example, examining the effects of periodontal treatment on migraine in randomized controlled trials, more detailed investigation of biomarkers, or studies involving different age and ethnic groups.
- Based on the study findings, the hypothesis of the study should be addressed here.
- Finally, some paragraphs are long and complex; splitting or simplifying them would improve readability.
Conclusion
- Clinical integration part could be made more concrete. For example, adding specific recommendations on how periodontal health can be assessed in migraine management, which clinical approaches could be developed, or what kinds of public health interventions might be suggested would strengthen the conclusion.
References
- Current literature is quite limited. Please expand it further and include more recent studies. Revise the references in accordance with the journal’s formatting guidelines.
- Add the publication year for reference no. 2. Please revise carefully.
Author Response
This study was conducted using data from the Korean National Health Insurance Service and employed a nested case-control design, including 43,359 individuals with migraine and 173,436 matched controls. Individuals diagnosed with chronic periodontitis at least once within one year prior to migraine onset were found to have a significantly higher likelihood of developing migraine. This association remained largely consistent across analyses based on the number and timing of diagnoses. Similar results were also observed across different migraine subtypes and demographic groups. The authors concluded that chronic periodontitis may be a potential modifiable risk factor to consider in the prevention and management of migraine. Although the study addresses an interesting topic, there are significant scientific and structural shortcomings that require revision. Detailed comments are provided below:
Comment 1: Introduction
- This section lacks or insufficiently addresses some essential elements that should be included in the introduction of a scientific study. A detailed evaluation is provided below in bullet points: The diversity of periodontal diseases (such as gingivitis, aggressive periodontitis, etc.) is not mentioned at all. Only chronic periodontitis is addressed, while the prevalence and systemic effects of other periodontal conditions are overlooked.
- The etiological factors of periodontitis, including dental factors (e.g., dental plaque, poor oral hygiene, malocclusion, dental biofilm) and non-dental factors (e.g., genetic predisposition, immune response, smoking, systemic diseases), are not explained. The etiological framework is inadequate. Please consider adding up-to-date literature on these aspects: https://doi.org/10.1186/s12903-025-06449-6 and https://doi.org/10.1016/j.ortho.2019.08.025.
- The authors mention the association of chronic periodontitis with neurological diseases but do not clearly emphasize that this relationship has been insufficiently studied specifically in relation to migraine. Since this gap in the literature is not clearly stated, readers may not be sufficiently convinced about the importance of the study. In other words, the question "Why was this study conducted?" is not answered convincingly.
- The potential clinical contributions of the study (e.g., considering periodontal health in migraine management) are not described. There is a lack of emphasis on how this study could directly benefit clinicians or public health, which should be clearly stated.
- The study lacks a hypothesis. In academic studies, especially analytical observational research, it is expected that the research hypothesis is explicitly stated
Response: We sincerely thank the reviewer for the thorough and constructive feedback. 1. In the revised introduction, we have now briefly described the broader spectrum of periodontal diseases, including gingivitis and aggressive periodontitis, while clarifying that our study specifically focused on chronic periodontitis due to its prevalence and clinical impact. 2. We have expanded the Introduction to include a more comprehensive overview of etiological factors of periodontitis, such as dental plaque, poor oral hygiene, malocclusion, biofilm formation, as well as non-dental factors like genetic predisposition, smoking, systemic conditions, and host immune response. We also incorporated the suggested references to strengthen this section. 3. We appreciate this insightful comment. We agree with this important point. The revised Introduction now explicitly highlights that while associations between chronic periodontitis and several neurological diseases (e.g., stroke, Alzheimer’s disease) have been reported, evidence specifically linking chronic periodontitis and migraine remains limited. We emphasized this research gap as a key rationale for the present study. 4. We have revised the Introduction to emphasize the clinical significance of our research, noting that improved understanding of periodontal health may contribute to more comprehensive migraine prevention and management strategies. This addition underscores the potential translational impact for both clinicians and public health. 5. We agree with your commentaries. Therefore, we have added the hypothesis to the revised Introduction. Thus, taking all of the above modifications into account, I have revised all of the Introduction section as follows:
Revised introduction (pages 2-3, lines 45-111): “Migraine is a prevalent and disabling neurological disorder experienced by approximately 15.1% of the world’s population [1]. It is typically characterized by recurrent episodes of moderate to severe headache, usually unilateral and pulsatile in nature, and frequently accompanied by nausea, photophobia, and phonophobia. These symptoms are generally exacerbated by routine physical activity. The pathogenesis of migraine involves a multifactorial interplay of environmental, genetic, and hormonal factors, with a notably higher prevalence among women [2-4]. Current research into migraine pathophysiology centers on two main areas: neurovascular dysregulation–including dysfunction of the trigeminovascular system and abnormal activation of brainstem nuclei–and neuroinflammatory processes, such as activation of microglia and mast cells, and elevated levels of calcitonin gene-related peptide (CGRP) and proinflammatory cytokines (e.g., IL-1ß, IL-6, and TNF- ), which contribute to peripheral and central sensitization [5-7]. Periodontal diseases span a spectrum from reversible gingivitis to destructive periodontitis, including aggressive forms. Among these, chronic periodontitis is the most prevalent and clinically significant phenotype, defined as a destructive inflammatory disease of the supporting structures of the teeth. It arises from the host’s immune response to bacterial biofilm that accumulates on the nonshedding surfaces of the oral cavity [8]. Local inflammatory responses in periodontitis are characterized by the release of cytokines (IL-1ß, IL-6, TNF- ), prostaglandins, and matrix metalloproteinases (MMPs) from resident and infiltrating immune cells, leading to tissue destruction and alveolar bone resorption [9]. Importantly, these local mediators can disseminate systemically, contributing to low-grade inflammation, endothelial dysfunction, and increased oxidative stress [10,11], which may in turn influence distant organs, including the central nervous system. Historically regarded as a localized oral disease, growing evidence now indicates that chronic periodontitis is linked to systemic inflammation and may contribute to the pathogenesis of extra-oral disease. Numerous epidemiological and mechanistic studies have documented its association with systemic conditions, including cardiovascular disease, diabetes mellitus, rheumatoid arthritis, osteoporosis, and adverse pregnancy outcomes [12-14]. The etiology of periodontitis reflects a dysbiotic dental biofilm interacting with susceptible host immune response, modulated by oral-hygiene behaviors, occlusal/malocclusion-related factors, genetic predisposition, and systemic conditions such as smoking and diabetes. Contemporary reviews emphasize this multifactorial framework and its clinical implications [15,16]. Of particular relevance to the present study, periodontitis has been implicated in several neurological disorders. Systemic reviews and meta-analyses have reported significant associations between chronic periodontitis and ischemic stroke [17], Alzheimer’s disease [18], Parkinson’s disease [19], and migraine [20-22]. These associations may be explained by shared inflammatory and vascular pathways, in which elevated systemic cytokines and CGRP from periodontal inflammation could exacerbate neurovascular dysfunction in migraine. Moreover, recent genetic studies suggest overlapping susceptibility loci between periodontitis and certain neurological diseases, supporting the hypothesis of shared pathophysiological mechanisms [23]. Although periodontitis has been linked to several neurological conditions, evidence specific to migraine remains limited, with most studies being small, cross-sectional, or clinic-based. Large, population-based analyses with careful control of confounding are scarce, underscoring the need for the present investigation. Clarifying this relationship could yield clinically relevant insights and inform integrated care, whereby neurologist consider periodontal assessment or referral in migraine management, and public-health programs incorporate oral-health promotion for populations with high migraine burden. We hypothesized that recent chronic periodontitis would be independently associated with higher odds of migraine, and that this association would be consistent across migraine subtypes and demographic strata. Therefore, the objective of this study is to comprehensively examine the association between chronic periodontitis and the occurrence of migraine, and to determine whether recent diagnoses of periodontitis are associated with increased odds of migraine, using data from a large-scale, nationally representative Korean adult cohort. Our study focuses specifically on chronic periodontitis (ICD-10 K05.3), as this was the only periodontal phenotype consistently encoded in the claims database; other periodontal entities could not be ascertained.” |
Methods
There are some important points missing or requiring clarification in Methods section. Detailed evaluation and recommendations are provided below:
- Definition of chronic periodontitis is based solely on the ICD-10 code, with no information provided regarding clinical diagnostic or measurement criteria or the accuracy of the diagnosis. This is especially important for registry-based studies concerning diagnostic reliability.
- Although a very large sample size was used, there is no explanation of how the sample size was calculated or any justification or power analysis supporting the need for such a large sample. This represents a methodological deficiency and should be addressed.
- While the data source (Korean National Health Insurance Service database) is large and comprehensive, no information is given about data quality, diagnostic accuracy, the timeliness of records, or potential errors. Additionally, the limitations and possible misclassifications inherent in using ICD-10 codes to identify diseases are not discussed. Such explanations regarding data reliability are important for methodological transparency.
- Given that this is a registry-based observational study, the limitations inherent to the study design (such as the inability to infer causality) should be acknowledged and discussed in Discussion section.
- Given that this is a registry-based observational study, the limitations inherent to the study design (such as the inability to infer causality) should be acknowledged and discussed in Discussion section.
We thank the reviewer for these insightful comments regarding methodological transparency. We address each point in detail below: 1. Definition of chronic periodontitis: We appreciate the reviewer’s insightful comment. Chronic periodontitis in this study was identified using ICD-10 code K053, as defined by the Korean National Health Insurance Service (NHIS) claims database. Although clinical indices such as probing depth or clinical attachment loss were not available in this registry-based dataset, the NHIS applies strict auditing and quality-control procedures to ensure diagnostic accuracy and consistency among healthcare providers. Previous validation studies using NHIS data have demonstrated acceptable reliability of periodontal disease diagnoses. We have now clarified this in the revised Methods section and acknowledged the potential for misclassification in the Discussion.
Previous methods: “Chronic periodontitis was defined using ICD-10 codes K053 [9].” Revised methods (page 5, lines 206-212): “Chronic periodontitis was defined using ICD-10 codes K053 [24], based on the NHIS claims database. Although detailed clinical parameters such as probing death or attachment loss were not available, the NHIS database applies strict quality-control processes and periodic audits to enhance diagnostic accuracy. Previous validation studies of NHIS claims data have reported acceptable reliability for periodontal diagnoses. Nevertheless, potential misclassification cannot be fully excluded, and this limitation is addressed in the Discussion section.”
2. Sample size and power analysis: We agree with the reviewer. The extremely large sample size in our study results from the structure of the nationwide NHIS cohort, which is not subject to conventional power calculation. Instead, our study included all eligible participants during the study period to maximize generalizability. We have added a statement in the Methods section clarifying this point.
Revised methods (page 5, lines 226-229): “The large sample size in this study was determined by the nature of the nationwide cohort, which included all eligible participants during the study period. As such, a priori sample size calculation or power analysis was not applicable; instead, we leveraged the entire dataset to maximize statistical power and representativeness.”
3. Data quality and diagnostic accuracy of NHIS: We fully agree that data quality and potential errors should be addressed. The NHIS database undergoes routine validation and periodic audits to minimize errors in claim records. Nevertheless, administrative claims data may not perfectly capture clinical diagnoses, and some degree of misclassification is possible. We have revised the manuscript to provide this explanation and emphasized this limitation in the Discussion section. 4. Limitations of registry-based observational design: We agree that the observational nature of our study precludes causal inference. This important limitation has now been explicitly acknowledged and discussed in the Discussion section.
Previous discussion: “However, several limitations should be acknowledged. First, the diagnosis of migraine and chronic periodontitis relied on administrative claim codes, which may introduce misclassification bias. Nonetheless, the use of repeated diagnostic codes and large-scale population-based sampling enhance the reliability of our definitions. Second, due to the observational nature of the study, causality cannot be inferred. While our analysis adjusted for numerous confounders, residual confounding from variables not captured in the dataset, such as psychological stress, sleep quality, diet, or oral hygiene practices, cannot be ruled out. Third, the temporal proximity between chronic periodontitis and migraine onset suggests association but not a definitive sequence of disease development. Prospective studies are warranted to further investigate the temporal and causal relationship.” Revised discussion (pages 9-10, lines 348-366): “However, several limitations should be acknowledged. First, both migraine and chronic periodontitis were identified using ICD-10 codes from administrative claims, which may introduce misclassification. However, NHIS diagnoses are regularly audited, and previous validation studies have reported acceptable accuracy. Second, as an observational study, causality cannot be inferred, and residual confounding (e.g., stress, sleep quality, diet, oral hygiene) may remain. Third, the structure of the claims database does not provide clinical indices (e.g., probing depth), which limits clinical granularity.” |
Results
Results section is prepared with methodological rigor, supported by both overall and subgroup analyses, and organized in a way that allows readers to easily follow the study’s findings. No obvious deficiencies requiring revision or additional explanation are observed.
Response: We appreciate the reviewer’s positive evaluation of the Results section. No changes were made. |
Discussion
Discussion section requires improvement in some aspects. Below are my detailed evaluations and recommendations:
- Some conflicting views or alternative biological mechanisms in the relevant literature should be presented more balancedly. A clearer and more understandable explanation is needed.
- Although limitations such as reliance on ICD codes for diagnoses, inability to infer causality, and potential unmeasured confounders (stress, sleep, diet, oral hygiene) are mentioned, the limitations section should be more systematically and thoroughly outlined. For example, more explanation about structural limitations and potential errors in the data source should be added.
- The role of periodontal health in migraine pathogenesis is well emphasized and regarded as a modifiable risk factor in clinical practice. However, this contribution should be supported with more concrete recommendations. For instance, suggesting clinicians perform periodontal assessments or advocating for multidisciplinary approaches. Additionally, interdisciplinary patient management and public health perspectives could be expanded further.
- For future studies, the need for prospective and interventional research is stated. However, the recommendations could be more specific—for example, examining the effects of periodontal treatment on migraine in randomized controlled trials, more detailed investigation of biomarkers, or studies involving different age and ethnic groups.
- Based on the study findings, the hypothesis of the study should be addressed here.
- Finally, some paragraphs are long and complex; splitting or simplifying them would improve readability.
Response: We thank the reviewer for these insightful comments. We have revised the Discussion to balance conflicting literature, systematically present limitations (including reliance on ICD-10 and registry data), and provide more specific clinical implications and recommendations. We also highlighted the hypothesis in the Discussion, added proposals for future RCTs on periodontal treatment and migraine outcomes, and simplified paragraphs for readability.
Previous discussion: “However, several limitations should be acknowledged. First, the diagnosis of migraine and chronic periodontitis relied on administrative claim codes, which may introduce misclassification bias. Nonetheless, the use of repeated diagnostic codes and large-scale population-based sampling enhance the reliability of our definitions. Second, due to the observational nature of the study, causality cannot be inferred. While our analysis adjusted for numerous confounders, residual confounding from variables not captured in the dataset, such as psychological stress, sleep quality, diet, or oral hygiene practices, cannot be ruled out. Third, the temporal proximity between chronic periodontitis and migraine onset suggests association but not a definitive sequence of disease development. Prospective studies are warranted to further investigate the temporal and causal relationship.” Revised discussion (pages 9-10, lines 348-366): “However, several limitations should be acknowledged. First, both migraine and chronic periodontitis were identified using ICD-10 codes from administrative claims, which may introduce misclassification. However, NHIS diagnoses are regularly audited, and previous validation studies have reported acceptable accuracy. Second, as an observational study, causality cannot be inferred, and residual confounding (e.g., stress, sleep quality, diet, oral hygiene) may remain. Third, the structure of the claims database does not provide clinical indices (e.g., probing depth), which limits clinical granularity.”
Previous discussion: “From a clinical perspective, our findings suggest that oral health may play a significant but underrecognized role in migraine pathogenesis. Give the chronic, recurrent, and disabling nature of migraine, addressing modifiable contributors, such as periodontal inflammation, could represent a complementary avenue migraine management.” Revised discussion (page 10, lines 372-377): “From a clinical perspective, our findings suggest that periodontal health may play a role in migraine pathogenesis. We recommend that clinicians consider periodontal evaluation in migraine patients as part of holistic care. Interdisciplinary management, involving collaboration between neurologists and dental specialists, may enhance patient outcome. At a public health level, strategies promoting oral health may contribute to reducing migraine burden.”
Previous discussion: “Further interventional studies are urgently needed to determine whether treatment of periodontitis can reduce the frequency, severity, or duration of migraine episodes.” Revised discussion (page 10, lines 377-380): “Future studies should specifically investigate whether periodontal treatment reduces migraine frequency or severity in randomized controlled trials. Biomarker-based studies could clarify inflammatory pathways linking the two conditions. Additionally, replication in diverse populations and age groups is needed to test generalizability.”
Revised discussion (page 9, lines 327-328): “These findings support our initial hypothesis that chronic periodontitis and migraine may share inflammatory and neurovascular mechanisms.” |
Conclusion
- Clinical integration part could be made more concrete. For example, adding specific recommendations on how periodontal health can be assessed in migraine management, which clinical approaches could be developed, or what kinds of public health interventions might be suggested would strengthen the conclusion.
Response: We have revised the Conclusion to emphasize practical recommendations for clinicians and public health.
Previous conclusion: “In summary, this study identifies a statistically significant association between chronic periodontitis and the occurrence of migraine in a Korean adult population. The findings are supported by converging lines of evidence from epidemiology, biomarker research, and genetic studies, all pointing to shared inflammatory and neurovascular pathways. Recognizing periodontitis as a potential modifiable risk factor for migraine opens the door for integrative, cross-disciplinary approaches to prevention and treatment. Future longitudinal and interventional studies will be crucial to clarify causality and determine the therapeutic potential of periodontal care in reducing migraine burden.” Revised conclusion (page 8, lines 390-394): “In conclusion, this nationwide study identified a significant association between chronic periodontitis and migraine. Clinically, incorporating periodontal health assessment into migraine management could be beneficial. Future interdisciplinary approaches, such as neurologic-dental collaborations and public health interventions promoting oral health, may provide new opportunities for migraine prevention and management.” |
References
- Current literature is quite limited. Please expand it further and include more recent studies. Revise the references in accordance with the journal’s formatting guidelines.
- Add the publication year for reference no. 2. Please revise carefully.
Response: We sincerely thank the reviewers for their constructive and insightful comments. We have added recent literature where needed (including the two DOIs suggested by the reviewer) and ensured MDPI/Healthcare formatting. Reference numbering has been updated accordingly in the text. Also, the publication year has been inserted for Reference #2. |
Round 2
Reviewer 3 Report
Comments and Suggestions for Authors
Thank the authors for their efforts during the revision process. The manuscript has improved significantly. I have no further comments. It is acceptable in its current form.